# Risk factors for vaginal mesh erosion after sacrocolpopexy in Korean women

**Tae Yeon Kim[1], Myung Jae Jeon**[1,2]*

**1** Department of Obstetrics and Gynecology, Seoul National University Hospital, Seoul, Korea, **2** Department of Obstetrics and Gynecology, Seoul National University College of Medicine, Seoul, Korea

* jeonmj@snu.ac.kr

**Data Availability Statement:** All relevant data are within the paper and its Supporting Information files.

**Funding:** The authors received no specific funding for this work.

## Abstract

### Objective

Although sacrocolpopexy (SCP) can provide durable apical support, the use of mesh may give rise to various complications, including vaginal mesh erosion. The aim of this study was to identify the risk factors for vaginal mesh erosion after SCP in Korean women.

### Methods

This retrospective cohort study included 363 women who underwent SCP with type 1 polypropylene mesh. They were evaluated at 1, 4, and 12 months after surgery and then annually thereafter with respect to anatomy and complications. Univariate and multivariate analyses using the Cox proportional hazard model were performed to identify the risk factors for mesh erosion.

### Results

During the median 2-year follow-up period, vaginal mesh erosion was found in 29 women (8.0%). Among them, 19 (65.5%) required surgical correction. Estrogenic status was the only independent risk factor for mesh erosion. The risk for mesh erosion was 4.5 times higher in premenopausal women than in menopausal women not on estrogen replacement therapy (ERT) (95% confidence intervals [CI] 1.9–10.9, p<0.01). Menopausal women on ERT also had an increased risk, with a statistically marginal significance (hazard ratio 2.5, 95% CI 0.9–6.6; p = 0.07).

### Conclusions

Premenopausal or menopausal women on ERT are at high risk for mesh erosion after SCP with type 1 polypropylene mesh, and two-thirds of mesh erosion cases require reoperation. This information should be incorporated into patient counseling and treatment decisions.

**Competing interests:** The authors have declared that no competing interests exist.

## Introduction

Resuspension of the vaginal apex is an essential component of reconstructive surgery for pelvic organ prolapse (POP) [1]. Loss of vaginal apical support is almost always present in cases of advanced POP [2], and surgical correction of the anterior and posterior walls may fail unless the apex is adequately supported because of the significant contribution of the vaginal apex to anterior and posterior vaginal support [3, 4]. A variety of procedures are available to correct vaginal apical prolapse; among them, sacrocolpopexy (SCP) is considered the gold standard [5].

SCP is the procedure that suspends the upper vagina from the sacral promontory with a synthetic mesh, and it can be performed via laparotomy or laparoscopy (with or without robotic assistance). Although this procedure can provide more durable apical support than other surgical options using native tissue, the use of mesh may give rise to various complications [6, 7]. One of the most commonly reported mesh-related complications is vaginal mesh erosion. A systematic review of SCP conducted in 2004 reported a 3.4% overall occurrence rate of mesh erosion [8]. However, most of the studies included in that review had a short-term follow-up duration, and a recent long-term follow-up study has demonstrated that the rate of mesh erosion increases over time, with an estimated probability of 10% by 7 years, and two-thirds of mesh erosion cases require surgical correction [9]. Therefore, vaginal mesh erosion is an issue that should be included in the preoperative decision-making process, and the risk factors for this complication should be identified.

Several previous studies have reported that smoking, concomitant hysterectomy, estrogen replacement therapy (ERT), and an advanced stage of POP might increase the risk of vaginal mesh erosion after SCP [10–15]. However, these studies had some methodologic limitations (e.g. a small sample size, the use of non-type 1 polypropylene mesh [not currently used because of the risk of infection], and possible confounding effects resulting from the diversity of surgeons). Moreover, the majority of women included in these studies were Caucasian; therefore, the results cannot be directly applied to women of different ethnic backgrounds.

The aim of this study was to identify the risk factors for vaginal mesh erosion after SCP with type 1 polypropylene mesh in Korean women.

## Materials and methods

### Patient and data collection

After obtaining approval from the institutional review board (SNUH 1907-045-1046) for this retrospective cohort study, we reviewed the medical records of 366 patients who had undergone SCP for symptomatic POP at the Seoul National University Hospital between November 2008 and June 2018. Of these patients, 3 women who did not attend any follow-up visits were excluded from the analysis.

All examinations and operative procedures were performed by one urogynecological subspecialist (M.J. Jeon). At baseline, demographic and medical history data were collected during an interview, and a standardized Pelvic Organ Prolapse Quantification (POPQ) examination was performed in a 45° upright sitting position with an empty bladder [16]. SCP was performed in a manner similar to the techniques described in a previous report [17]. In brief, after opening the vesicovaginal and rectovaginal space, the peritoneum over the sacral promontory was incised vertically, and the peritoneal incision was extended to the cul-de-sac. A 10-cm × 4-cm polypropylene mesh (Gynemesh PS; Ethicon, Somerville, NJ) fashioned in a Y shape from two pieces of mesh was used to secure the pubocervical and rectovaginal fascia. The anterior and posterior leaf of the mesh were secured to the proximal 3 cm of the vaginal cuff using 6 interrupted stitches of delayed absorbable suture (Polysorb 2–0; Covidien PLC,

Dublin, Ireland). The proximal arm of the Y-shaped mesh was then secured to the anterior longitudinal ligament of the sacrum using either 2 or 3 interrupted stitches of non-absorbable suture (Prolene 0; Ethicon). Then, the mesh was retroperitonealized using interrupted stitches of delayed absorbable sutures (Vicryl 2–0; Ethicon). Women with a uterus in situ underwent concomitant total hysterectomy, and posterior repair and a transobturator tape procedure were performed as indicated.

Scheduled in-person follow-ups occurred at 1, 4, and 12 months and then annually thereafter. At each visit, patients underwent a POPQ examination and speculum examination to screen for vaginal mesh erosion. In addition, symptoms related to mesh erosion were assessed.

## Statistical analysis

Data were analyzed with SPSS software (version 25; SPSS Inc., Chicago, IL). Univariate and multivariate analyses using the Cox proportional hazard model were conducted to identify risk factors for vaginal mesh erosion. The mesh erosion rates were estimated with the use of the Kaplan-Meier method. A p-value of <0.05 was considered statistically significant.

## Results

Table 1 displays the baseline characteristics of the study population. The median age was 65 (interquartile range, 54–76) years, and most women were menopausal and not on ERT. There were no current smokers, and 90% presented advanced POP (POPQ stage 3 or 4). SCP was

**Table 1. Characteristics of the study population (n = 363).**

| Variables | Value |
|---|---|
| Age at surgery, yr | 65.0 (54.0–76.0) |
| Vaginal parity | 3.0 (1.0–5.0) |
| Body mass index, kg/m$^2$ | 24.5 (20.9–28.1) |
| Estrogenic status | |
| Premenopausal | 36 (9.9) |
| Menopausal on ERT | 55 (15.2) |
| Menopausal not on ERT | 272 (74.9) |
| Current smoker | 0 |
| Hypertension | 165 (45.5) |
| Diabetes mellitus | 56 (15.4) |
| Prior hysterectomy | 100 (27.5) |
| Prior prolapse surgery | 45 (12.4) |
| POPQ stage | |
| 2 | 38 (10.5) |
| 3–4 | 325 (89.5) |
| Route of SCP | |
| Open | 309 (85.1) |
| Laparoscopic | 54 (14.9) |
| Concomitant procedures | |
| Total hysterectomy | 263 (72.5) |
| Posterior repair | 158 (43.5) |
| Midurethral slings | 157 (43.3) |
| Postoperative sexual activity | 99 (27.3) |

Values are presented as the median (interquartile range) or number (%).

ERT = estrogen replacement therapy; POPQ = pelvic organ prolapse quantification; SCP = sacrocolpopexy

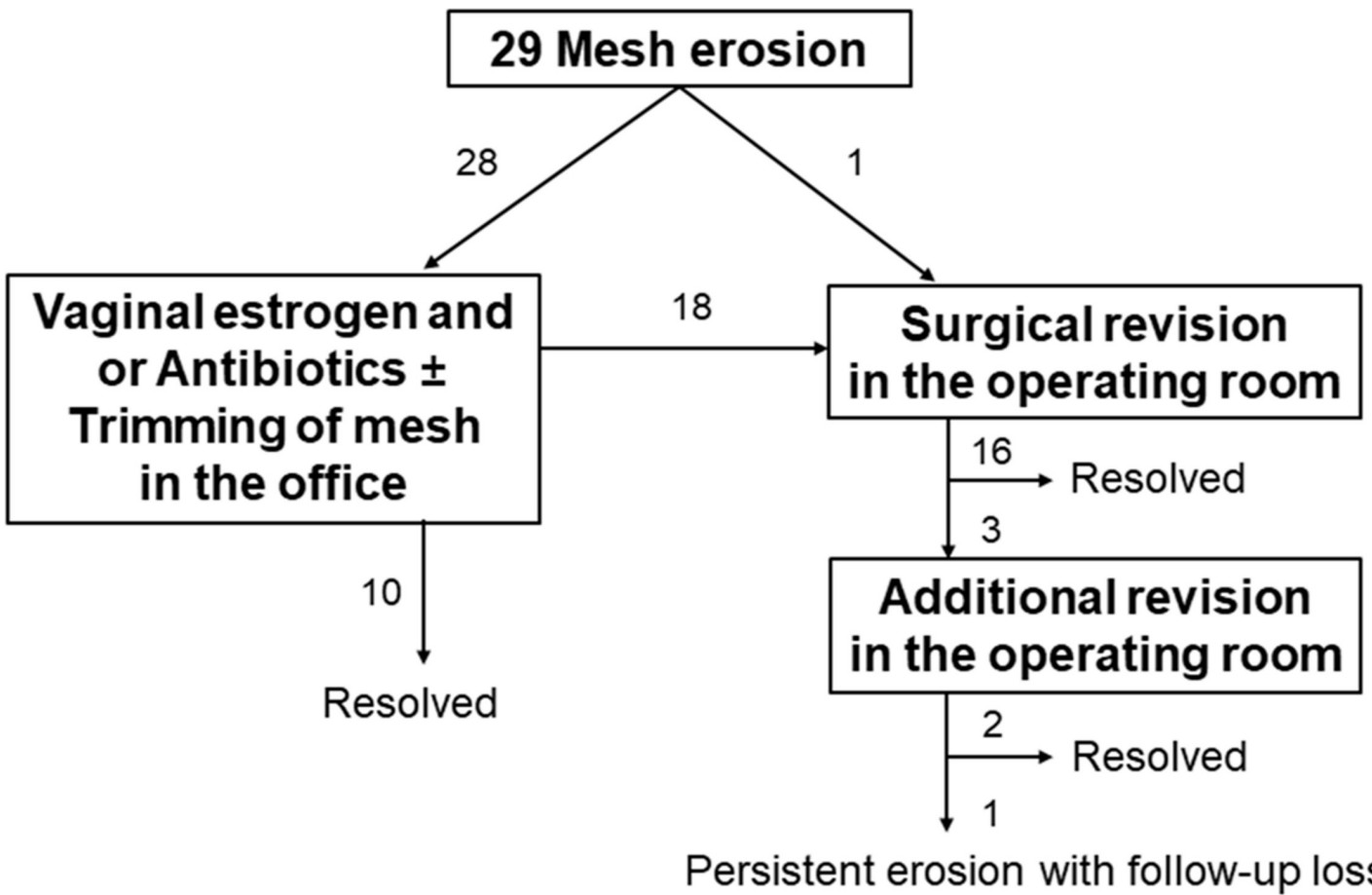

**Fig 1. Outcomes of the management of vaginal mesh erosion after sacrocolpopexy.**

mostly performed by open surgery (85%), and 73% of the patients underwent concomitant total hysterectomy. Twenty-seven percent of the patients were sexually active after surgery.

During the median 2-year follow-up period (range, 1–116 months), vaginal mesh erosion was found in 29 women (8.0%). All erosions were located in the vaginal apex. The median interval from surgery to mesh erosion detection was 4 months (range, 1–56 months). The most frequent complaint was abnormal vaginal bleeding or discharge (55%); however, the remainder (45%) were asymptomatic and mesh erosion was observed during speculum examination. Of the 29 mesh erosions, 28 were initially treated by conservative therapy consisting of vaginal estrogen and/or antibiotics with or without trimming of the exposed mesh in the office and 1 by surgical revision in the operating room. Eighteen mesh erosions (64%) did not resolve with conservative therapy and finally required surgical revision in the operating room (Fig 1).

Univariate and multivariate analyses with the Cox proportional hazard model revealed that estrogenic status was the only independent risk factor for mesh erosion. The risk for mesh erosion was 4.5 times higher in premenopausal women than in menopausal women not on ERT (95% confidence intervals [CI] 1.9–10.9, p<0.01). Menopausal women on ERT also had an increased risk, with a statistically marginal significance (hazard ratio 2.5, 95% CI 0.9–6.6; p = 0.07) (Table 2). The cumulated 2-year mesh erosion rates in premenopausal women, menopausal women on ERT, and menopausal women not on ERT were 21%, 10%, and 5%, respectively (Fig 2).

**Table 2. Risk factors for vaginal mesh erosion after sacrocolpopexy.**

| Variable | Univariate | | Multivariate[a] | |
|---|---|---|---|---|
| | HR | 95% CI | HR | 95% CI |
| Vaginal parity | 0.82 | 0.59–1.15 | | |
| Body mass index, kg/m$^2$ | 0.94 | 0.83–1.07 | | |
| Estrogenic status | | | | |
| Menopausal not on ERT | 1.00 | (reference) | 1.00 | (reference) |
| Premenopausal | 6.41 | 2.83–14.54 | 4.49 | 1.86–10.87 |
| Menopausal on ERT | 2.54 | 0.95–6.76 | 2.48 | 0.93–6.62 |
| Hypertension[b] | 0.37 | 0.16–0.87 | 0.53 | 0.22–1.28 |
| Diabetes mellitus[b] | 0.67 | 0.20–2.20 | | |
| Prior prolapse surgery[b] | 0.88 | 0.27–2.89 | | |
| POPQ stage | | | | |
| 2 | 1.00 | (reference) | 1.00 | (reference) |
| 3–4 | 0.33 | 0.14–0.77 | 0.50 | 0.21–1.23 |
| Route of SCP | | | | |
| Open | 1.00 | (reference) | | |
| Laparoscopic | 1.09 | 0.38–3.14 | | |
| Surgeon's experience, per 10 cases | 0.98 | 0.95–1.02 | | |
| Concomitant total hysterectomy[b] | 1.33 | 0.54–3.27 | | |
| Concomitant posterior repair[b] | 1.21 | 0.58–2.50 | | |
| Concomitant transobturator tape[b] | 0.68 | 0.32–1.47 | | |
| Postoperative sexual activity[b] | 1.81 | 0.87–3.76 | | |

CI = confidence interval; ERT = estrogen replacement therapy; HR = hazard ratio; POPQ = pelvic organ prolapse quantification; SCP = sacrocolpopexy

[a]Performed with variables of significant values from univariate analysis (p<0.05).

[b]Present versus absent

## Discussion

Our study shows that estrogenic status is associated with an increased risk of vaginal mesh erosion after SCP with type 1 polypropylene mesh in Korean women. The risk for mesh erosion was 4.5 times higher in premenopausal women than in menopausal women not on ERT. Menopausal women on ERT also had a 2.5-fold risk, with a statistically marginal significance. In addition, two-thirds of mesh erosion cases did not respond to conservative therapy and finally required surgical correction in the operating room.

Given the high recurrence rates after native tissue repairs and the US Food and Drug Administration warnings on transvaginal mesh, SCP has increasingly been used to correct apical vaginal prolapse [18]. However, SCP also has the potential for mesh-related complications, and vaginal mesh erosion has been reported in up to 27% of patients [19]. Previously, several studies have investigated the risk factors for vaginal mesh erosion after SCP. Although smoking, concomitant hysterectomy, ERT, and an advanced stage of POP have been suggested as risk factors, none have been consistently shown to increase the mesh erosion rates [10–15, 20, 21]. Our study found that estrogenic status could increase the risk of vaginal mesh erosion after SCP.

Considering the beneficial effect of estrogen on extracellular matrix metabolism in pelvic supportive tissues [22–24], this finding may be surprising. Although the reason for the association between estrogenic status and higher rates of vaginal mesh erosion is not clear, some plausible explanations are as follows. First, premenopausal and menopausal women on ERT are relatively young to develop POP; in Korea, POP surgeries are most often performed on women 70 years and older [25]. Younger patients with POP are more likely to have a genetic

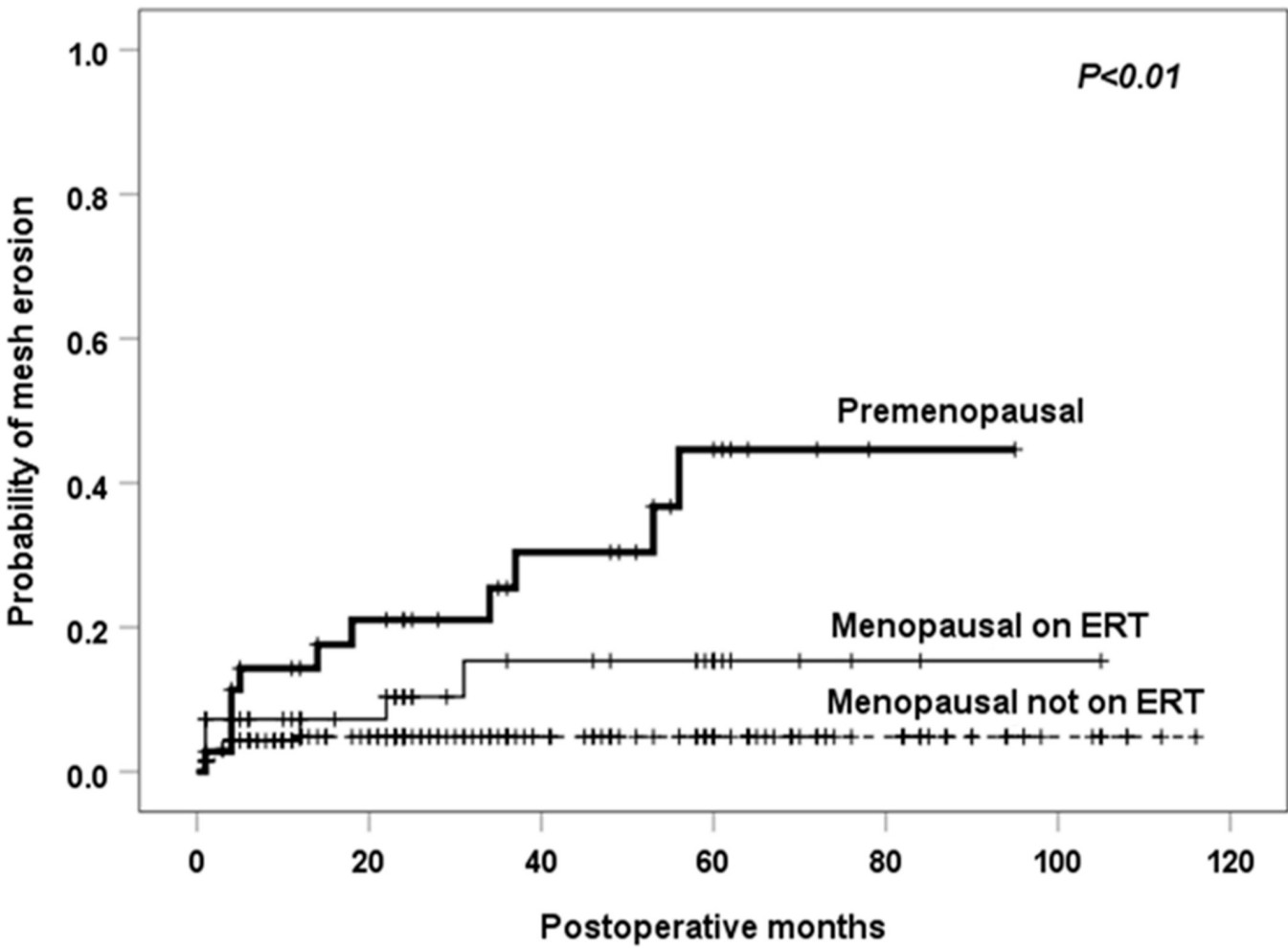

**Fig 2. Kaplan-Meier failure curve for vaginal mesh erosion after sacrocolpopexy according to estrogen status.** ERT = estrogen replacement therapy. The p-value was calculated using the log-rank test based on all available follow-up data.

predisposition to impaired connective tissue regeneration, thus altering the wound healing process in the presence of synthetic mesh [26]. Second, ERT could have been started for vaginal atrophy even though the actual indication for ERT could not be determined due to the retrospective design of this study. Therefore, menopausal women on ERT were more susceptible to mesh erosion due to a thinner and atrophic vaginal epithelium. Third, estrogen itself might have a negative impact on the wound healing process in the presence of synthetic mesh. Gynemesh PS is known to induce strong foreign body inflammatory responses to the mesh insertion site, and prolonged activation of matrix metalloproteinases secreted by inflammatory cells can destroy collagen and elastin [27]. In contrast to the positive effects on the vaginal epithelium, estrogen impacts the injured stroma by decreasing collagen and elastin synthesis and the expression of growth factors and anti-inflammatory cytokine [28]. Therefore, estrogen can hinder the wound healing of a grafted vagina, which may also explain the finding by Wu et al. that concomitant hysterectomy was associated with mesh erosion after SCP only in women on ERT [12].

Our study has several strengths. First, this is the first study to investigate the risk factors for vaginal mesh erosion after SCP in an Asian population. Although our findings may not be

applicable to women of other ethnic backgrounds, a recent systematic review that included various races also demonstrated that premenopause or ERT is a significant risk factor for mesh erosion after female pelvic floor reconstructive surgery [29]. Second, all surgeries were performed with type 1 polypropylene mesh by a single expert surgeon, which minimizes the possible confounding effects of different surgical techniques or skill levels and mesh type on the outcome. Third, the analysis of time-to-event outcomes minimized attrition bias, selection bias due to follow-up loss.

Nonetheless, there were some limitations, mainly attributable to the inherent weaknesses of a retrospective study. Another limitation is the relatively small sample size and number of events, which might be not sufficient to evaluate some potential risk factors. Smoking is also frequently reported to be a significant risk factor for mesh erosion [10, 11]; however, there were no smokers in our study population, and we could not evaluate the relationship between smoking and mesh erosion. The use of Gynemesh PS might have also affected our results. Although lighter meshes have been developed, Gynemesh PS was the only type 1 polypropylene mesh available in Korea during the study period. After implantation, Gynemesh PS can induce stronger foreign body inflammatory responses than lighter type 1 polypropylene mesh, and this may influence the occurrence of mesh erosion [27]. Nonetheless, two retrospective cohort studies comparing the mesh erosion rates after SCP with use of ultra-lightweight versus heavier-weight polypropylene mesh showed conflicting results, with an earlier recurrence observed in the ultra-lightweight mesh group [30, 31]. Well-designed studies will be required to clarify this issue.

## Conclusion

Premenopausal or menopausal women on ERT are at high risk for mesh erosion after SCP with type 1 polypropylene mesh, and two-thirds of mesh erosion cases require reoperation. This information should be incorporated into patient counseling and treatment decisions.

## Supporting information

**S1 File. The database of this manuscript.**
(SAV)

## Author Contributions

**Conceptualization:** Myung Jae Jeon.

**Data curation:** Tae Yeon Kim.

**Formal analysis:** Myung Jae Jeon.

**Investigation:** Tae Yeon Kim, Myung Jae Jeon.

**Supervision:** Myung Jae Jeon.

**Writing – original draft:** Tae Yeon Kim.

**Writing – review & editing:** Myung Jae Jeon.

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
