## [Decision Letter · Decision Letter 0]

5 Dec 2019

PONE-D-19-27662

Risk factors for vaginal mesh erosion after sacrocolpopexy in Korean women

PLOS ONE

Dear Dr. Jeon,

Thank you for submitting your manuscript to PLOS ONE. After careful consideration, we feel that it has merit but does not fully meet PLOS ONE’s publication criteria as it currently stands. Therefore, we invite you to submit a revised version of the manuscript that addresses the points raised during the review process.

We would appreciate receiving your revised manuscript by Jan 19 2020 11:59PM. To enhance the reproducibility of your results, we recommend that if applicable you deposit your laboratory protocols in protocols.io, where a protocol can be assigned its own identifier (DOI) such that it can be cited independently in the future. For instructions see: http://journals.plos.org/plosone/s/submission-guidelines#loc-laboratory-protocols

We look forward to receiving your revised manuscript.

Kind regards,

Antonio Simone Laganà, M.D.

Academic Editor

PLOS ONE

Journal Requirements:

1. Please include captions for your Supporting Information files at the end of your manuscript, and update any in-text citations to match accordingly. Please see our Supporting Information guidelines for more information: http://journals.plos.org/plosone/s/supporting-information.

Additional Editor Comments (if provided):

The topic of the manuscript is interesting. Nevertheless, the reviewers raised several concerns: considering this point, I invite authors to perform the required major revisions.

Reviewers' comments:

Reviewer's Responses to Questions

**Comments to the Author**

1. Is the manuscript technically sound, and do the data support the conclusions?

Reviewer #1: Partly

2. Has the statistical analysis been performed appropriately and rigorously? 

Reviewer #1: No

3. Have the authors made all data underlying the findings in their manuscript fully available?

Reviewer #1: Yes

4. Is the manuscript presented in an intelligible fashion and written in standard English?

Reviewer #1: No

5. Review Comments to the Author

Reviewer #1: I was pleased to revise the manuscript entitled “Risk factors for vaginal mesh erosion after sacrocolpopexy in Korean women” (Manuscript Number: PONE-D-19-27662).

The study approval was viewed by the institutional review board of Seoul National University Hospital (No. 1907-045-1046).

In general, this manuscript was aimed to identify risk factor for mesh erosion in women underwent sacrocolpopexy type 1 polypropylene mesh. In my honest opinion, the topic is interesting enough to attract the readers’ attention.

Methodology requires to be improved, although conclusions are supported by the reported data. Nevertheless, authors should clarify some point and improve the discussion citing relevant and novel key articles about the topic.

In general, the Manuscript may benefit from several major revisions, as suggested below:

1. I would suggest a revision of the manuscript language to improve some typos and readability in some sections.

2. Abstract. The statement “in spite of the use of type 1 polypropylene mesh” cannot be supported by the study results because all women underwent correction with the same mesh.

3. The statistical methods are not clear. How was the erosion rate estimated using Kaplan-Meier curves? If it refers to the hazard ration of erosion, it is more appropriate the cox analysis. I would suggest checking this section and improving description.

4. Were some specific inclusion and exclusion criteria used?

5. Table 2. I would suggest reporting the line with HR of 1 indicating the group of reference. Regarding the surgeon experience, does it include the firs surgical procedure?

6. Based on the possible age factor related to estrogen exposure as risk factor for erosion, I would suggest test the age as confounding factor in the multivariate analysis.

7. Lines 165-166. I would suggest use the appropriate term for the described bias, such as attrition bias. Moreover, the use of cox analysis cannot identify differences that may be linked to risk of erosion and the lost at follow-up. The main way is to evaluate if the characteristics of patients lost at follow-up are comparable with followed up women.

8. Discussion. I would suggest discussing, at least briefly, the key role of a multidisciplinary approach for the management of patients with pelvic organ prolapse (DOI: 10.5114/pm.2019.89496).

9. As highlighted by the study results, 8% of women developed mesh erosion and 68% of them underwent surgery. In addition to the treatment of mesh erosion, the recurrence of pelvic organ prolapses may be a complex procedure. Indeed, although several approaches are able for the management of POP, the best strategy in case of recurrence after vaginal vault prolapse still remains debated. I would discuss, at least briefly, the available evidence about novel techniques, referring to: PMID: 29675427; PMID: 29038834.

6. PLOS authors have the option to publish the peer review history of their article (what does this mean?). If published, this will include your full peer review and any attached files.

Reviewer #1: No

---

## [Author Response · Author response to Decision Letter 0]

16 Dec 2019

Dear Antonio Simone Laganà, Academic Editor of PLOS ONE,

Thank you very much for the review of our manuscript. The comments were excellent and helpful, and were good guides for revising and improving our manuscript.

The following is an itemized account of the changes in the manuscript made in response to the comments.

Journal Requirements:

1. Please include captions for your Supporting Information files at the end of your manuscript, and update any in-text citations to match accordingly. Please see our Supporting Information guidelines for more information: http://journals.plos.org/plosone/s/supporting-information.

-> We added caption for our supporting information file at the end of the manuscript.

Reviewers' comments:

Reviewer #1: I was pleased to revise the manuscript entitled “Risk factors for vaginal mesh erosion after sacrocolpopexy in Korean women” (Manuscript Number: PONE-D-19-27662).

The study approval was viewed by the institutional review board of Seoul National University Hospital (No. 1907-045-1046).

In general, this manuscript was aimed to identify risk factor for mesh erosion in women underwent sacrocolpopexy type 1 polypropylene mesh. In my honest opinion, the topic is interesting enough to attract the readers’ attention.

Methodology requires to be improved, although conclusions are supported by the reported data. Nevertheless, authors should clarify some point and improve the discussion citing relevant and novel key articles about the topic.

In general, the Manuscript may benefit from several major revisions, as suggested below:

1. I would suggest a revision of the manuscript language to improve some typos and readability in some sections.

-> We edited English.

2. Abstract. The statement “in spite of the use of type 1 polypropylene mesh” cannot be supported by the study results because all women underwent correction with the same mesh.

-> We revised the sentence.

3. The statistical methods are not clear. How was the erosion rate estimated using Kaplan-Meier curves? If it refers to the hazard ration of erosion, it is more appropriate the cox analysis. I would suggest checking this section and improving description.

-> Kaplan-Meier survival analysis presents the estimated cumulative erosion rates as well as survival curve. Therefore, we can obtain the estimated erosion rates with the use of the Kaplan-Meier method.

4. Were some specific inclusion and exclusion criteria used?

-> No. We included all patients who had undergone SCP for symptomatic POP at the Seoul National University Hospital between November 2008 and June 2018 except for 3 women who did not attend any follow-up visit.

5. Table 2. I would suggest reporting the line with HR of 1 indicating the group of reference. Regarding the surgeon experience, does it include the first surgical procedure?

-> We revised Table 2 according to your recommendation.

We included the whole SCP to evaluate the effect of surgeon’s experience on the mesh erosion (even though the surgeon completed urogynecologic fellowship training course and acquired skill for SCP).

6. Based on the possible age factor related to estrogen exposure as risk factor for erosion, I would suggest test the age as confounding factor in the multivariate analysis.

-> We intentionally excluded age from the multivariate analysis because of the close relationship between age and estrogenic status (younger women are more likely to premenopausal and take ERT to control their vasomotor symptoms). Statisticians recommended us not including age in the multivariate analysis.

 In addition, when we included both age and estrogenic status in the multivariate analysis, estrogenic status, not age, was an independent risk factor for vaginal mesh erosion.

7. Lines 165-166. I would suggest use the appropriate term for the described bias, such as attrition bias. Moreover, the use of cox analysis cannot identify differences that may be linked to risk of erosion and the lost at follow-up. The main way is to evaluate if the characteristics of patients lost at follow-up are comparable with followed up women.

-> We revised the sentence.

There are various reasons for follow-up loss: from no specific discomfort to disappointment to the surgical outcomes (recurrence of POP or complications). To avoid attribution bias, we can contact the patients by phone. However, mesh erosion does not always cause problems such as vaginal bleeding, discharge, and dyspareunia. Therefore, patients should be examined in the office to evaluate the occurrence of mesh erosion. In addition, mesh erosion rates increase over time, and there is no acceptable cut-off time to evaluate the occurrence of mesh erosion. We think that the use of Cox proportional hazard model is the best way to evaluate the risk factors for vaginal mesh erosion making the most efficient use of the data in this retrospective study.

8. Discussion. I would suggest discussing, at least briefly, the key role of a multidisciplinary approach for the management of patients with pelvic organ prolapse (DOI: 10.5114/pm.2019.89496).

-> We carefully read the article entitled to “Multidisciplinary management of women with pelvic organ prolapse, urinary incontinence and lower urinary tract symptoms. A clinical and psychological overview”. That article deals with a multidisciplinary approach for female sexual dysfunction, which is beside the point of our manuscript (The aim of our study was to evaluate the risk factors for vaginal mesh erosion after SCP).

9. As highlighted by the study results, 8% of women developed mesh erosion and 68% of them underwent surgery. In addition to the treatment of mesh erosion, the recurrence of pelvic organ prolapses may be a complex procedure. Indeed, although several approaches are able for the management of POP, the best strategy in case of recurrence after vaginal vault prolapse still remains debated. I would discuss, at least briefly, the available evidence about novel techniques, referring to: PMID: 29675427; PMID: 29038834.

 -> We agree with your opinion. The use of mesh can reduce the recurrence of POP while it can give rise to various complications including mesh erosion. Nonetheless, current scientific evidences support that SCP using type 1 polypropylene mesh is appropriate for women with risk factors for prolapse recurrence. Our study found that premenopausal and postmenopausal women, known at high risk for the recurrence of POP, have increased risk for vaginal mesh erosion. We would like to highlight this point.

 We read the articles (recommended by you) entitled to “Transvaginal Bilateral Sacrospinous Fixation after Second Recurrence of Vaginal Vault Prolapse: Efficacy and Impact on Quality of Life and Sexuality” and “The treatment of post-hysterectomy vaginal vault prolapse: a systematic review and meta-analysis”. The latter article is in accord with our manuscript, but it seems better to cite it in the introduction section rather than discussion section (because the aim of that study was to compare the effectiveness and safety among various vaginal vault suspension procedures, rather than to evaluate the risk factors for mesh erosion after SCP). We cited that article as reference 7 in the introduction section.

---

## [Decision Letter · Decision Letter 1]

21 Jan 2020

Risk factors for vaginal mesh erosion after sacrocolpopexy in Korean women

PONE-D-19-27662R1

Dear Dr. Jeon,

We are pleased to inform you that your manuscript has been judged scientifically suitable for publication and will be formally accepted for publication once it complies with all outstanding technical requirements.

With kind regards,

Antonio Simone Laganà, M.D., Ph.D.

Academic Editor

PLOS ONE

Additional Editor Comments (optional):

Authors performed the required corrections, which were positively evaluated by the reviewer. I am pleased to accept this paper for publication.

Consider that the appropriate term for the bias is “attrition” and not “attribution”. Please change it or leave just selection bias.

Reviewers' comments:

Reviewer's Responses to Questions

**Comments to the Author**

1. If the authors have adequately addressed your comments raised in a previous round of review and you feel that this manuscript is now acceptable for publication, you may indicate that here to bypass the “Comments to the Author” section, enter your conflict of interest statement in the “Confidential to Editor” section, and submit your "Accept" recommendation.

Reviewer #1: All comments have been addressed

2. Is the manuscript technically sound, and do the data support the conclusions?

Reviewer #1: Yes

3. Has the statistical analysis been performed appropriately and rigorously? 

Reviewer #1: Yes

4. Have the authors made all data underlying the findings in their manuscript fully available?

Reviewer #1: Yes

5. Is the manuscript presented in an intelligible fashion and written in standard English?

Reviewer #1: Yes

6. Review Comments to the Author

Reviewer #1: I was pleased to revise the manuscript entitled “Risk factors for vaginal mesh erosion after sacrocolpopexy in Korean women” (Manuscript Number: PONE-D-19-27662).

The study approval was viewed by the institutional review board of Seoul National University Hospital (No. 1907-045-1046).

In general, this manuscript was aimed to identify risk factor for mesh erosion in women underwent sacrocolpopexy type 1 polypropylene mesh. In my honest opinion, the topic is interesting enough to attract the readers’ attention.

The authors addressed almost all the suggested revisions, and I appreciated the manuscript improvement.

The appropriate term for the bias is “attrition” and not “attribution”. Please change it or leave just selection bias.

7. PLOS authors have the option to publish the peer review history of their article (what does this mean?). If published, this will include your full peer review and any attached files.

Reviewer #1: No

---

## [Editor Report · Acceptance letter]

23 Jan 2020

PONE-D-19-27662R1 

Risk factors for vaginal mesh erosion after sacrocolpopexy in Korean women 

Dear Dr. Jeon:

I am pleased to inform you that your manuscript has been deemed suitable for publication in PLOS ONE. Congratulations! Your manuscript is now with our production department. 

With kind regards,

on behalf of

Dr. Antonio Simone Laganà 

Academic Editor

PLOS ONE